

# Motor performance in Prader-Willi syndrome patients and its potential influence on caregiver's quality of life

Valeria Jia-Yi Chiu[1], Li-Ping Tsai[2,3], Jang-Ting Wei[1], I-Shiang Tzeng[4] and Hsin-Chi Wu[1,3]

[1] Department of Rehabilitation Medicine, Taipei Tzu Chi Hospital, Buddhist Tzu Chi Medical Foundation, New Taipei City, Taiwan

[2] Department of Pediatrics, Taipei Tzu Chi Hospital, Buddhist Tzu Chi Medical Foundation, New Taipei City, Taiwan

[3] Department of Medicine, Tzu Chi University, Hualien, Taiwan

[4] Department of Research, Taipei Tzu Chi Hospital, Buddhist Tzu Chi Medical Foundation, New Taipei City, Taiwan

Corresponding author
Hsin-Chi Wu,
hsinchiwu@tzuchi.com.tw

## ABSTRACT

**Background.** Prader-Willi syndrome (PWS) is a complex, multisystem genetic disorder characterized by a variety of physical, cognitive, and behavioral impairments. PWS is a unique sarcopenia model characterized by an abnormal increase in body fat mass and a decrease in muscle mass that predisposes patients to reduced physical activity, functional limitations, and disability. These manifestations may require both symptomatic and supportive management, thus negatively influencing their lifelong family caregiver's quality of life. The aim of this study was to examine the functional motor performance of adults with PWS in Taiwan and to measure the quality of life of their primary family caregivers.

**Methods.** The functional motor tests consisted of the following: (1) 30-s sit-to-stand test, (2) timed up-and-go test, (3) hand grip and lateral pinch strength tests, and (4) Berg Balance Scale. The World Health Organization Quality of Life-short form (WHOQOL-BREF) and the Short-Form 36 Health Survey Questionnaire (SF-36) were used to evaluate health-related quality of life, and the parenting stress index was used to assess the magnitude of stress within the parent-child system.

**Results.** The participants included seven adults (two females and five males) with genetically confirmed PWS and their respective main caregivers. The mean age of the adults with PWS was 25.28 years; range 18–31 years, SD 5.10; the mean BMI was 29.2 kg/m$^2$, SD 6.43. All adults with PWS showed lower hand grip and lateral pinch strengths, fewer sit-to-stand cycles during the 30-s chair stand test, and greater average time during the timed up-and-go test when compared to the normative data on healthy adults. Balance was negatively correlated with the caregiver's health concepts of social functioning ($r_s$ −0.879, $P = 0.009$) and with role limitations due to physical problems ($r_s$ −0.899, $P = 0.006$) and emotional problems ($r_s$ −0.794, $P = 0.033$); hand grip strength was negatively correlated with bodily pain ($r_s$ −0.800, $P = 0.031$), as assessed using the SF-36 questionnaire. The timed up-and-go test was positively correlated with the social relationship domain ($r_s$ 0.831, $P = 0.021$), as assessed using the WHOQOL-BREF questionnaire. The parenting stress index showed no association with the PWS patient's physical activities.
**Conclusions**. All adults with PWS showed decreased upper and lower limb strength and functional mobility when compared to healthy adults. Some of their motor performance might have negative effects on their primary family members in terms of social participation and physical and emotional role limitations. Future research should explore the relationship between physical performances, psychological difficulties of PWS and caregiver's QOL.

## INTRODUCTION

Prader–Willi syndrome (PWS) is a multisystem genetic disorder caused by a loss of paternally inherited expression on the chromosome 15q11-13. Causes of this disruption can be attributed to maternal uniparental disomy of chromosome 15 (mUPD), a deletion (DEL) of a 5–6 Mb region from paternally contributed chromosome 15, or imprinting defects (*Prader, Labhart & Willi, 1956*; *Goldstone, 2004*; *Cassidy et al., 2012*). The prevalence of PWS is estimated to be 1/10,000–1/30,000 (*Cassidy et al., 2012*). A newborn with PWS may present with severe hypotonia and poor sucking ability, followed by global developmental delays in the future. There could also be facial anomalies, such as almond-shaped eyes and thin upper lips, acromicria (small hands and feet), and genital hypoplasia, which should alert clinicians to conduct further genetic testing (*Goldstone et al., 2008*). Seven different nutritional stages were identified in the majority of children with PWS; in subphase 1a, the child with PWS is hypotonic without obesity but having problems with feeding, which is sometimes accompanied by failure to thrive; when the infant starts to grow in subphase 1b, weight increases at a normal rate; in subphase 2a, weight increases but caloric intake is maintained; when the child has a median age of 4.5 years in subphase 2b, interest in food and weight gain starts; hyperphagia is characterized in phase 3 when the child reaches a median age of eight years, and it continues throughout adulthood during which food seeking and a lack of sense of satiety are prominent; some adults progress to phase 4 when there is no longer an insatiable appetite, and the individual is able to feel full (*Cassidy et al., 2012*). Children with PWS may also encounter psychological and behavioral problems, such as compulsivity, temper tantrums, stubbornness, attention deficit and hyperactivity disorders, and even autism, that have been reported to worsen with age and diminish in older adults, interfering with their quality of life (QOL) during adolescence and adulthood (*Elena et al., 2012*; *Cassidy et al., 2012*).

Patients with PWS suffer multiple physical, developmental, and behavioral issues that require their families to devote time and effort to care for them (*Mazaheri et al., 2013*). These manifestations may require both symptomatic and supportive management (*McCandless & Committee on Genetics, 2011*). The QOL of patients with PWS and its relationship with the clinical picture (*Caliandro et al., 2007*) and temper outbursts (*Tunnicliffe et al., 2014*) has been studied. *Hodapp, Dykens & Masino (1997)* used the Freidrich-Stress Questionnaire and found higher levels of parental stress and

pessimism in the families of children with PWS compared with the families of children with intellectual disabilities of mixed etiologies. Multiple studies have indicated that parents and caregivers are at an increased risk of experiencing emotional distress, depression and anxiety when a child or young adult has health concerns (*Curfs & Fryns, 1992*; *Sarimski, 1997*; *Van Lieshout et al., 1998*). However, most published studies focus on the QOL of parents of children with PWS and are related to the childrens' psychological symptoms, such as stress, depression, and social isolation. Little attention has been paid to adults with PWS or to their lifelong family caregivers. There was only one study concerned with the QOL of caregivers, which might deteriorate during the late adolescence of patients with a mUPD genotype. However, the reason for this finding remains unclear (*Ihara et al., 2014*).

When compared with obese non-PWS, abnormal body composition with an increase in body fat mass and a 25–37% decrease in muscle mass were noted in patients with PWS (*Reus et al., 2011*), representing a unique congenital model of sarcopenia (*Irizarry et al., 2016*). This could be the result of a hormonal deficiency as a consequence of hypothalamic dysfunction. Patients with PWS also display structural and functional muscle abnormalities and hypo-excitable cortical motor areas (*Reus et al., 2011*). Decreased muscle strength and hypotonia predisposes to significantly reduced physical activity, functional limitations, and disability, mainly due to lower energy utilization (*Verbrugge & Jette, 1994*; *Rantanen et al., 1999*; *Butler et al., 2007*). PWS patients can also have abnormal gait patterns and score well below the normal ranges on standardized functional motor tests (*Cimolin et al., 2010*). The physical performance deficit of PWS is significant and could be a source of burden for caregivers of PWS patients. However, no studies have shown a relationship between PWS's motor problems and the lifelong family caregiver's QOL. Due to the paucity of research, we conducted this study to assess the functional motor performance of adults with PWS and its potentially negative impact on their main family caregiver's QOL.

## MATERIALS AND METHODS

### Participants

Seven adults with genetically confirmed PWS (aged 18 years or older) and their family caregivers were recruited through the Prader-Willi Syndrome Association (Taiwan). The PWS participants and their main caregivers needed to have a sufficient command of the Mandarin language to understand the study information and the questionnaires. The participants had to give their informed and written consent to participate. The study followed the principles of the Declaration of Helsinki and was approved by the Research Ethics Committee of Buddhist Taipei Tzu Chi General Hospital (03-XD45-082).

### Procedure

Examinations at both the impairment level (e.g., muscle strength) and the functional mobility level (e.g., ambulation) are recognized as important aspects of the decision-making process involved in managing daily activities. The physical performance battery included the following five tests that were used in the sample of adults with PWS: two upper limb muscle strength tests (hand grip and lateral pinch), two lower body muscle

strength, walking speed and endurance function tests (the timed up-and-go test and the 30-s chair stand test), and one balance test (the Berg Balance Scale).

Versions of the World Health Organization Quality of Life-short form (WHOQOL-BREF) (*Yao et al., 2002*), the Short-Form 36 Health Survey Questionnaire (SF-36) (*Lu, Tseng & Tsai, 2003*), and the parenting stress index (PSI) (*Yeh et al., 2001*) validated for the Taiwanese population were administered to the PWS patients' main caregivers to determine their global QOL, health-related QOL, and the magnitude of stress in the parent and child relationship, respectively.

Handgrip and lateral pinch strength were assessed in the dominant hand in pounds-force using a Baseline hydraulic dynamometer and pinch gauge, respectively. Three 3-s sustained hand grips and lateral pinches were obtained, with a 1-min interval between each measurement. The highest of the three handgrip and lateral pinch measurements were included in the analysis.

The 30-s chair stand test is an important lower body strength clinical test. Poor functions related to getting up from a chair or climbing stairs are related to decreasing levels of the demands of daily activities (*Hughes, Myers & Schenkman, 1996*; *Jones, Rikli & Beam, 1999*). In addition, this test assesses endurance by counting the number of sit to stand ups achieved and has also been used to monitor training and rehabilitation (*Millor et al., 2013*).

The timed up-and-go (TUG) test measures the time a PWS patient requires to "rise from a chair, walk a 3-meter-long line on the floor, turn around, walk back, and sit down again" (*Podsiadlo & Richardson, 1991*), all of which are critical for independent mobility. Adults without balance problems can complete the test in less than 10 s, but those with limited mobility skills take more than 30 s to complete the task (*Mahoney & Barthel, 1965*). The TUG test has been correlated with the Berg Balance Scale (*Bennie et al., 2003*), gait speed/time (*Freter & Fruchter, 2000*), and stair climbing (*Hughes, Osman & Woods, 1998*).

Static and dynamic activities of varying difficulty were assessed using the Berg Balance Scale (BBS). The items include simple mobility tasks (e.g., transferring, standing unsupported, sitting-to-standing) and more difficult tasks (e.g., turning to look behind, turning 360°, reaching forward with an outstretched arm, tandem standing, single-leg stand) (*Berg et al., 1989*).

The WHOQOL-BREF measures global QOL. It comprises four domains related to quality of life (physical health, psychological health, social relationships, and environment) and one facet measuring overall quality of life and general health. It contains 28 items comprising 26 standard items and two national (culturally relevant) items. The two national Taiwanese version items are as follows: "Do you feel respected by others?" in the social relationships domain, and "Are you usually able to have the things you like to eat?" in the environmental domain. Scores range from 4 to 20 (*Yao et al., 2002*).

The SF-36 assesses health-related quality of life and contains 36 items in eight areas as follows: (1) limitations in physical activities because of health problems; (2) limitations in social activities because of physical or emotional problems; (3) limitations in usual role activities because of physical health problems; (4) bodily pain; (5) general mental health; (6) limitations in usual role activities because of emotional problems; (7) vitality

**Table 1  Characteristics of the individuals with Prader-Willi syndrome.**

| Age | Sex | Gene | Height (m) | Weight (kg) | BMI (kg/m$^2$) | GH Tx | Grip (lbs) | Pinch (lbs) | TUG (s) | S-S (times) | BBS/56 | Care-age |
|-----|-----|------|-----------|-------------|----------------|-------|------------|-------------|---------|-------------|--------|----------|
| 31 | M | UPD | 1.70 | 71 | 24.6 | + | 65 | 11 | 9 | 14 | 50 | 61 |
| 28 | M | U/K | 1.60 | 109 | 42.5 | + | 65 | 10 | 14 | 9 | 51 | 48 |
| 31 | M | Del | 1.51 | 67 | 28.9 | + | 55 | 12 | 17 | 9 | 44 | 58 |
| 24 | M | U/K | 1.65 | 76 | 27.9 | + | 31 | 8 | 13 | 7 | 49 | 54 |
| 19 | F | Del | 1.44 | 56 | 27 | + | 35 | 9 | 19 | 5 | 35 | 49 |
| 26 | F | Del | 1.52 | 53 | 22.9 | + | 40 | 9 | 10 | 10 | 53 | 56 |
| 19 | M | U/K | 1.53 | 73 | 31.1 | + | 60 | 12 | 9 | 10 | 48 | 47 |

Notes.
Gene, genetics; Del, microdeletion type; UPD, maternal uniparenteral disomy type; U/K, unknown; BMI, body mass index; TUG, timed up-and-go test; S-S, sit-to-stand; BBS, Berg Balance Scale.

(energy and fatigue); and (8) general health perception (*Lu, Tseng & Tsai, 2003*). These scales are scored from 0 to 100 following a standard algorithm, and higher scores represent better health.

The Parenting Stress Index (PSI) is a 101-item self-report measure that assesses an individual's degree or extent of stress in the role of a parent in caring for a child. There are two major domains in the Parenting Stress Index, the parent domain and the child domain (*Yeh et al., 2001*). The parent domain includes seven subscales, and the child domain includes six subscales. The child domain addresses characteristics that make it difficult for parents to fulfill their parenting roles. The parent domain measures sources of stress and potential dysfunction in the parent–child system that might be related to the dimensions of the parent's functioning (*Yeh et al., 2001*).

## Statistical methods

The data were analyzed using SPSS 20.0 (IBM SPSS Statistics). The results are presented as the mean (SD) or median (range). The correlation of the PWS patients' physical motor performance and their main caregiver's quality of life was assessed using Spearman's correlation test. A significance level of $p < 0.05$ was set for all tests. Spearman correlation coefficient ($r_s$) values of $+1$ indicate a perfect positive association of ranks; an $r_s$ of zero indicates no association between the ranks; and $r_s$ of $-1$ indicates a perfect negative association of ranks.

## RESULTS

Seven individuals (Two females and five males, mean age of 25.28 years; age range 18–31 years; SD 5.10; mean BMI of 29.2 kg/m$^2$; SD 6.43) with PWS and their respective main caregivers were recruited through the Prader-Willi Syndrome Association (Taiwan). All adults with PWS showed lower hand grip and lateral pinch strengths, a lesser number of sit-to-stand cycles during the 30-s chair stand test, and greater average time when performing the TUG test when compared to the healthy adult normative data. Details of the sample are given in Table 1.

**Table 2 Correlation between physical performance and SF-36 results.**

| SF-36 | Grip | | Pinch | | TUG | | Sit-stand | | BBS | |
|---|---|---|---|---|---|---|---|---|---|---|
| | $r_s$ | $p$ | $r_s$ | $p$ | $r_s$ | $p$ | $r_s$ | $p$ | $r_s$ | $p$ |
| PF | −0.055 | 0.907 | −0.056 | 0.906 | 0.385 | 0.393 | −0.583 | 0.169 | −0.546 | 0.205 |
| RP | −0.38 | 0.401 | 0.187 | 0.688 | 0.176 | 0.706 | −0.393 | 0.384 | **−0.899**\*\* | **0.006**\*\* |
| RE | −0.477 | 0.279 | 0.26 | 0.574 | 0.353 | 0.438 | −0.385 | 0.394 | **−0.794**\* | **0.033**\* |
| VT | −0.523 | 0.228 | −0.333 | 0.465 | 0.275 | 0.55 | −0.389 | 0.389 | −0.6 | 0.154 |
| MH | −0.482 | 0.274 | −0.257 | 0.578 | 0.245 | 0.596 | −0.385 | 0.393 | −0.667 | 0.102 |
| SF | −0.397 | 0.379 | 0.248 | 0.592 | 0.33 | 0.469 | −0.324 | 0.478 | **−0.879**\*\* | **0.009**\*\* |
| BP | −0.523 | 0.228 | −0.056 | 0.906 | 0.385 | 0.393 | −0.491 | 0.263 | −0.709 | 0.074 |
| GH | −0.716 | 0.071 | −0.056 | 0.906 | 0.642 | 0.12 | −0.639 | 0.122 | −0.746 | 0.054 |
| Pain | **−0.8**\* | **0.031**\* | −0.266 | 0.564 | 0.309 | 0.5 | −0.395 | 0.381 | −0.324 | 0.478 |

Notes.

PF, physical functioning; RP, role limitations due to physical problems; RE, role limitations due to emotional problems; VT, vitality; MH, mental health; SF, social functioning; BP, body pain; GH, general health; TUG, timed up-and-go test; BBS, Berg Balance Scale.

\*$p < 0.05$.

\*\*$p < 0.01$.

**Table 3 Correlation between physical performance and the WHOQOL-BREF questionnaire.**

| WHOQOL domains | Grip | | Pinch | | TUG | | Sit-stand | | BBS | |
|---|---|---|---|---|---|---|---|---|---|---|
| | $r_s$ | $p$ | $r_s$ | $p$ | $r_s$ | $p$ | $r_s$ | $p$ | $r_s$ | $p$ |
| Physical | −0.395 | 0.381 | −0.296 | 0.519 | −0.028 | 0.953 | −0.083 | 0.859 | −0.255 | 0.582 |
| Psychological | −0.182 | 0.696 | −0.229 | 0.621 | 0.218 | 0.638 | −0.119 | 0.799 | −0.288 | 0.531 |
| Social relation | −0.227 | 0.625 | −0.076 | 0.871 | **0.831**\* | **0.021**\* | −0.619 | 0.138 | −0.58 | 0.172 |
| Environmental | −0.073 | 0.877 | −0.009 | 0.984 | 0.164 | 0.726 | 0.018 | 0.969 | −0.234 | 0.613 |

Notes.

TUG, timed up-and-go test; BBS, Berg Balance Scale.

\*$p < 0.05$.

Table 2 shows the results of the Spearman correlation analysis between the PWS participants' physical motor performance and their main caregivers' SF-36 QoL variables. There was a significant high negative correlation between the PWS patients' hand grip strength and the caregivers' bodily pain ($r_s$ −0.800, $P = 0.031$). Regarding the PWS participants' balance, there was a high negative correlation with the main caregivers' health-related role limitations due to physical problems ($r_s$ −0.899, $P = 0.006$), emotional problems ($r_s$ −0.794, $P = 0.033$) and social functioning ($r_s$ −0.879, $P = 0.009$).

Table 3 shows the results of the Spearman correlation analysis between the PWS participants' physical motor performance and their main caregivers' WHOQOL-BREF and PSI QOL variables. The TUG test showed a significant high positive correlation with the PWS patients and their family members' social relationships ($r_s$ 0.831, $P = 0.021$). The higher the PWS participants' mobility condition was, the greater their family members' participation was in social relationships.

The results of the Spearman correlation analysis between the PWS participants' physical motor performance and their main caregivers' PSI QOL variables are shown in Table 4. The 101 items of the PSI focus on two domains as follows: the child's characteristics
**Table 4** Correlation between physical performance and PSI.

| PSI | Grip | | Pinch | | TUG | | Sit-stand | | BBS | |
|---|---|---|---|---|---|---|---|---|---|---|
| | $r_s$ | $p$ | $r_s$ | $p$ | $r_s$ | $p$ | $r_s$ | $p$ | $r_s$ | $p$ |
| Total score | 0.288 | 0.531 | 0.509 | 0.243 | 0.054 | 0.908 | −0.164 | 0.726 | −0.321 | 0.482 |
| Parent characteristic | 0.127 | 0.786 | 0.128 | 0.784 | 0.091 | 0.846 | −0.266 | 0.564 | 0.036 | 0.939 |
| Child characteristics | 0.450 | 0.310 | 0.600 | 0.154 | −0.144 | 0.758 | 0.055 | 0.908 | −0.179 | 0.702 |

and the parent's characteristics/family context. The child characteristics domain consists of the following six subscales: adaptability, distractibility/hyperactivity, demandingness, mood, acceptability, and parent reinforcement. The parent characteristics domain consists of a total domain score and seven subscales, including competence, isolation, health, role restriction, depression, and spouse. Neither the child characteristics nor the parent characteristics showed significant associations with the motor performance of the adult with PWS.

## DISCUSSION

Prader-Willi syndrome is a genetic disorder characterized by obesity and other multisystem clinical manifestations that encompass both physical and behavioral abnormalities (*Holm et al., 1993*). Many studies have focused on the correlations between phenotypic aspects and genetic findings and on the relationship between phenotype and the risk of cardiac and respiratory problems, fractures, sleep disorders, and scoliosis. However, patients' perception of their pathology and their family's burden are important issues in solving families' everyday problems and improving their QOL (*McCandless & Committee on Genetics, 2011*; *Mazaheri et al., 2013*; *Tunnicliffe et al., 2014*; *Caliandro et al., 2007*). The families of children with PWS experience higher levels of parental stress and more pessimism than the families of children with other intellectual disabilities (*Hodapp, Dykens & Masino, 1997*). Family caregivers of people with PWS and Angelman syndrome in Western Australia, regardless of the condition and age of their offspring, experience considerable levels of stress, over long periods of time. The initial diagnosis, insufficient time for personal matters, and tiredness or even exhaustion resulting from disrupted and insufficient sleep were among the most stressful situations (*Thomson et al., 2016*). Individuals with PWS have the prominent characteristics of obesity, hypotonia, and decreased muscle strength, which at severe levels, are disabling to mobility and exercise capacities. This is the first report to evaluate the motor performance of adults with PWS and assess its negative influence on their main family caregivers' QOL.

Patients with PWS present a unique congenital model of sarcopenia with an abnormal body composition increase in the body fat mass and decrease in the muscle mass (*Irizarry et al., 2016*; *Reus et al., 2011*). Sarcopenia is defined as low muscle function (by a walking speed or grip strength) in the presence of low muscle mass (*McKee et al., 2017*). Small hands and feet (acromicria) are often seen in PWS. Adults with PWS exhibit hand and foot lengths <25th centile for age, and women present hand and foot lengths <50th centile for their height (*Hudgins & Cassidy, 1991*). We evaluated muscle strength and mobility of

adults with PWS of different ages and sexes. Our data showed lower hand grip, lateral pinch strengths, slower TUG and sit-to-stand tests in all adults with PWS when compared to the normative data of healthy adults (*Mathiowetz et al., 1985*). Grip strength was significantly correlated to subject height and hand span, width, and length, with hand length being more related to strength than hand width or hand span (*Macdermid, Fehr & Lindsay, 2002*). A larger muscle mass could be anticipated with a larger body size, indicating a direct biomechanical relationship between a PWS patient's hand size and hand output (*Macdermid, Fehr & Lindsay, 2002*).

It has been documented that hand grip strength predicted functional limitations and disability and that low grip strength was associated with a greater likelihood of functional limitations (*Rantanen et al., 1999*). *Tong et al. (2002)* studied caregiver burden from the physical functioning perspective and found that physical functioning was decreased in the female caregivers of children with a physical disability, and this decrease was associated with the caregivers' pain severity and mood. The results of these two studies might be consistent with our findings that there was a high negative correlation between the PWS patients' hand grip strength and the caregivers' bodily pain. Based on our knowledge, the reasons that the adults with PWS impaired their caregiver's QOL remain unknown; the result seems to be a need for further investigation in terms of the relationship between upper limb muscle strength in patients with PWS and their caregiver's bodily pain.

Patients with PWS score well below the normal range on standardized motor performance tests, and their normal gait pattern is disturbed (*Cimolin et al., 2010*). This results in a reduced physical capacity to perform tasks and increases their main caregivers' risk of emotional stress (*Curfs & Fryns, 1992*; *Sarimski, 1997*; *Van Lieshout et al., 1998*). We found that the balance of adults with PWS was highly negatively correlated to their main caregivers' health concepts of role limitations due to physical and emotional problems and, therefore, to their social functioning. The role limitation due to physical problems subscale of the SF-36 identifies the degree to which physical health interferes with work or other daily activities. Detrimental health-related behaviors, such as inadequate sleep and rest or inability to visit a doctor when necessary may cause some caregivers to experience role limitations due to physical problems (*Ren et al., 1998*). The role limitation due to emotional problems subscale is sensitive to differences in mental health status (*Eker & Tüzün, 2004*). Regarding the social functioning subscale, both physical health, such as caring for a child with physical impairments for a whole day, and emotional problems interfere with normal social activities. Lower scores could result from depression, anxiety, or problems with behavior and emotion adjustment problems (*Eker & Tüzün, 2004*). Caregivers who provide care on a regular basis often experience greater levels of depression and anxiety and are more likely to experience psychiatric and physical health problems compared with caregivers who receive assistance with care (*Pearlin & Skaff, 1995*). The role of a caregiver for an individual with PWS covers a very wide range of activities, such as helping their child with instrumental daily activities such as doing housework, assisting them with transportation, shopping, handling finances, and managing medication, which can cause physical strain for their main caregivers (*Asghar, Robabeh & Sahel, 2009*).

A significant high positive correlation was found between the TUG test results of the adults with PWS and their main caregivers' social relationships. The TUG test measures the speed at which an individual performs during several functional maneuvers, which include standing up, walking, turning and sitting down. As the time to complete the timed up-and-go test decreases, there is an increase in the BBS or Barthel score, indicating an increase in function. A TUG test time fewer than 20 s indicates that an individual is independent for basic transfers (*Podsiadlo & Richardson, 1991*). All adults with PWS scored within the normal performance ranges in the TUG test. However, when compared to the TUG normative reference values of normal adults in their 20 s (8.42 ± 1.40 s) and 30 s (8.56 ± 1.23 s) (*Kear, Guck & McGaha, 2016*), adults with PWS all scored below of the normative ranges, which could concern their main caregivers during longer distance mobility tasks, such as outdoor mobility, leisure activities, using public transportation, or shopping.

In our study, PSI variables showed negligible associations with the motor functions of the adults with PWS. In contrast to reports from Western Australia (*Thomson et al., 2016*), most caregivers in our study experienced no significant parenting burden. This could be possible because caregivers could have gradually become accustomed to the adult with PWS's behavior and motor function over the years as they grew, making light of or learning from previous negative situations (*Thomson et al., 2016*).

It is evident that there are limitations in the current study. It is a cross-sectional study with a small sample size. The data are heterogeneous due to a lack of analysis of genotype and psychological status. This is also a single association study, which makes generalization of results difficult. Additional studies that recruit more participants from other genetic or endocrinology clinics and include a long-term follow up are warranted for a more precise development understanding. Second, the participants were not compared to other obesity patterns in terms of body composition, physical activities, and orthopedic characteristics. To assess the impact of motor performance of adults with PWS on their caregiver's QOL, an age- or weight-matched control group comparison would make observing changes more accurate. In addition, because measurement queries were constructed for various purposes, we did not consider using a composite score from each of the three quality of life measures. Multiple comparison correction was also not used in this study.

Knowing that motor performance might have negative effects on their main caregiver's QOL, in terms of role limitations due to physical and emotional problems and their social function, may help caregivers find a coping strategy. It is advised that service providers, family members, and peer support associations could also provide practical and emotional support.

## CONCLUSIONS

To our knowledge, this is the first study to explore the possible correlation of physical performance of PWS and QOL of their caregivers. All PWS participants have decreased lower and upper limb strength and decreased functional mobility compared to healthy adults. The results of our study indicate that the motor performance of adults with PWS might have negative effects on their family members in terms of social participation and

physical and emotional role limitations. In addition, it is necessary to acknowledge that multiple hypothesis testing provides clues for statistical inference in this study. Future research should explore the relationship between physical performance, psychological difficulties of PWS and caregiver's QOL.

## ACKNOWLEDGEMENTS

We thank the adults with Prader-Willi syndrome and their families who participated in this study and the Prader-Willi Syndrome Association (Taiwan), which assisted in recruitment and data collection.

### Funding
The study was supported by the Taipei Tzu Chi Hospital, Buddhist Tzu Chi Medical Foundation (No. TCRD-TPE-104-50). The funders had no role in study design, data collection and analysis, decision to publish, or preparation of the manuscript.

### Grant Disclosures
The following grant information was disclosed by the authors:
Taipei Tzu Chi Hospital, Buddhist Tzu Chi Medical Foundation: TCRD-TPE-104-50.

### Competing Interests
The authors declare there are no competing interests.

### Author Contributions
- Valeria Jia-Yi Chiu performed the experiments, analyzed the data, wrote the paper, prepared figures and/or tables.
- Li-Ping Tsai conceived and designed the experiments, reviewed drafts of the paper, patients' referral.
- Jang-Ting Wei performed the experiments, contributed reagents/materials/analysis tools, wrote the paper, prepared figures and/or tables.
- I-Shiang Tzeng analyzed the data, wrote the paper, reviewed drafts of the paper, statistical analysis.
- Hsin-Chi Wu conceived and designed the experiments, reviewed drafts of the paper.

### Human Ethics
The following information was supplied relating to ethical approvals (i.e., approving body and any reference numbers):
The study was approved by the Research Ethics Committee of Buddhist Taipei Tzu Chi General Hospital (03-XD45-082).

### Data Availability
The raw data is contained in the tables.

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
