# Peer review of "Motor performance in Prader-Willi syndrome patients and its potential influence on caregiver's quality of life"

_PeerJ, doi:10.7717/peerj.4097_

## Round 0.1 · original submission · Major Revisions

· Academic Editor

Major Revisions

Please see the thoughtful comments from the two reviewers. It is absolutely critical that the statistical analysis be performed in a way that accounts for multiple comparisons. It is likely that some associations will no longer be significant once adjusted. The authors should make certain to reframe their discussion to highlight significant differences, to discuss the significant limitations, and to propose appropriate next steps. As suggested by both reviewers, the authors should make certain that the introduction of Prader-Willi syndrome and its clinical characteristics are accurate and that the specific knowledge gap in this population is discussed in relation to the literature on PWS.

Please make sure to address all of reviewers concerns in a point-by-point response.

·

Basic reporting

The majority of the manuscript is clear and unambiguous. I have a few suggestions on grammar and language.
1. Please use "patient first" language throughout the manuscript. For example, line 54 should read "the child with Prader-Willi" instead if "the Prader-Willi child".
2. The correct term is "hand grip strength", not "hand gripping strength" (example, line 35).
3. Line 54: The statement that excessive eating begins in infancy is incorrect. Increased interest in food has a median age of onset of 4.5 years and hyperphagia 8 years (Miller et al., American Journal of Medical Genetics, 2011). In infancy most patients with PWS have poor feeding, with or without failure to thrive.
4. line 64 - change to "with an increase in body fat mass"
5. line 73 - change to "devote time and effort to care for them"

Experimental design

1.The researchers do not present an a priori hypothesis. They aimed to investigate the relationship between motor performance and primary caregiver QOL. There were 80 correlations examined in the analysis section. Was there a particular relationship that was of interest? Or was this an hypothesis generating exercise?
2. The question is interesting and attempts to address a knowledge gap.
4. The patient population was adequately characterized.
5. Validated measures were used.

Validity of the findings

1. The researchers analyzed 80 correlation coefficients and did not correct for multiple hypothesis testing or specify and a priori hypothesis/primary outcome. It is not surprising that 5 of 80 results had a p-value <0.05 as this approximates the expected false positive error rate. The conclusions presented in the discussion vastly overstate the meaning of these results. For example, you could perhaps conclude that these results justify a subsequent study to test the relationship between upper limb muscle strength in patients with PWS and their caregiver's bodily pain. It is not appropriate to conclude that better upper limb muscle strength in PWS lowers the intensity of caregiver's pain (lines 215-218).
2. Lines 240-246 -The authors state that a TUG test time <20 seconds indicates that an individual is independent for basic transfers. All PWS patients in this study had a TUG time <20 seconds so how can we draw any conclusions between the TUG test results and caregiver social relationships?
3. I suggest using a composite score from each of the 3 quality of life measures, rather than 16 subscales. If that's not possible, the conclusion should acknowledge the multiple hypothesis testing and need for further research rather than making strong conclusions based on these data.

·

Basic reporting

The authors address an important topic to Prader-Willi, namely, the impact of adult patients’ physical disabilities on their family/caregiver quality of life

 There are a number of errors in grammar and syntax noted. As such, the manuscript would benefit from extensive professional and/or peer editing.

 The introduction introduces the phenotype and disabilities of PWS, however misstates several particulars. On line 54, the authors state that during infancy the Prader-Willi child starts excessive eating. In reality the hyperphagia of PWS usually starts in school age children, not infancy (Genetics in Medicine, 2012, 14(1):10-26). In addition, PWS is not cause by the deletion of a “specific gene” as the authors state on line 59, but a specific gene region which includes a number of potential genes leading to the phenotype.

 I would like to see the authors expand upon the knowledge gap being filled and be more explicit in stating their hypothesis and why this is important to know.

Experimental design

 Authors state on line 166 that seven individuals were recruited but on line 167 only describe 2 females and 4 males. Table 1 shows 5 males and 2 females.

 The authors state that all patients and their caregivers were recruited through the PWS Association. I am concerned that this may introduce significant recruitment bias as those who participate in such associations may be more engaged and adjusted to the difficulties of caring for family members with PWS. This bias combined with the small numbers in this study severely limit the applicability of these findings to the general PWS population. I would very much like to see additional study participants recruited through other means such as genetics or endocrinology clinic. Minimally, the authors should address this shortcoming in the Discussion.

 The authors do a good job of outlining the various physical and QoL tests performed as part of the study.

 I would like the authors to comment on the use of handgrip and lateral pinch strength tests in the PWS populations. Have these been validated as tests of upper limb muscle strength in PWS? Given the atypical hand morphology of PWS (brachydactyly with interdigital webbing and joint laxity) could these tests be interrogating hand dexterity more than upper limb strength in these patients?

Validity of the findings

 In lines 210-218, the authors discuss the correlation between grip strength and caregivers’ pain as assessed by the SF-26 results. I would like the authors to expand on their discussion regarding these results. Why was there no correlation with the Pinch test? What domain of upper limb strength or manual dexterity is the Grip test assaying that the Pinch test is not? Are these results expected or not? Why or why not?

 The authors correctly identify the limitations of this study in lines 255-260. These limitations severely curtail the general applicability of these results. Not only are the study numbers extremely small but there is no control group of any type. Are there similar studies that have been carried out in similar populations that the authors can contrast their results with? Other obese populations? Other disabled populations?

 If these results were to be validated, what specific changes in patient care might be made to help mitigate these detrimental effects on caregiver’s QoL?

---

## Round 0.2 · accepted · Accept

· Academic Editor

Accept

Thank you for your attention to the reviewers comments.